ecology

chronic wasting disease, selective harvesting, reindeer, Norway, adaptive management

**Author for correspondence:**
Atle Mysterud
e-mail: atle.mysterud@ibv.uio.no

# Harvest strategies for the elimination of low prevalence wildlife diseases

Atle Mysterud[1], Hildegunn Viljugrein[1,2], Christer M. Rolandsen[3] and Aniruddha V. Belsare[4,5]

[1]Centre for Ecological and Evolutionary Synthesis (CEES), Department of Biosciences, University of Oslo, PO Box 1066, Blindern, 0316 Oslo, Norway
[2]Norwegian Veterinary Institute, PO Box 750 Sentrum, 0106 Oslo, Norway
[3]Norwegian Institute for Nature Research (NINA), PO Box 5685 Torgarden, 7485 Trondheim, Norway
[4]Boone and Crockett Quantitative Wildlife Center, Department of Fisheries and Wildlife, Michigan State University, East Lansing, MI, USA
[5]Department of Biology, Emory College of Arts and Sciences, Emory University, Atlanta, GA, USA

 AM, 0000-0001-8993-7382; HV, 0000-0002-3798-5267;
CMR, 0000-0002-5628-0385; AVB, 0000-0002-4651-0116

The intensive harvesting of hosts is often the only practicable strategy for controlling emerging wildlife diseases. Several harvesting approaches have been explored theoretically with the objective of lowering transmission rates, decreasing the transmission period or specifically targeting spatial disease clusters or high-risk demographic groups. Here, we present a novel model-based approach to evaluate alternative harvest regimes, in terms of demographic composition and rates, intended to increase the probability to remove all infected individuals in the population during the early phase of an outbreak. We tested the utility of the method for the elimination of chronic wasting disease based on empirical data for reindeer (*Rangifer tarandus*) in Norway, in populations with (Nordfjella) and without (Hardangervidda) knowledge about exact disease prevalence and population abundance. Low and medium harvest intensities were unsuccessful in eliminating the disease, even at low prevalence. High-intensity harvesting had a high likelihood of eliminating the disease, but probability was strongly influenced by the disease prevalence. We suggest that the uncertainty about disease prevalence can be mitigated by using an adaptive management approach: forecast from models after each harvest season with updated data, derive prevalence estimates and forecast further harvesting. We identified the problems arising from disease surveillance with large fluctuations in harvesting pressure and hence sample sizes. The elimination method may be suitable for pathogens that cause long-lasting infections and with slow epidemic growth, but the method should only be attempted if there is a low risk of

reinfection, either by a new disease introduction event (e.g. dispersing hosts) or due to environmental reservoirs. Our simulations highlighted the short time window when such a strategy is likely to be successful before approaching near complete eradication of the population.

# 1. Introduction

Harvesting is the legal and regulated hunting of game species, and it represents the main cause of mortality among ungulates in Europe and North America [1]. Extensive harvesting of hosts by marksmen, often termed culling, comprises a controversial component of the management toolbox in the combat of infectious diseases in wildlife [2,3]. The ability of harvesting to control disease epidemics depends on a number of factors [4], and several mechanisms may be involved. For diseases with density-dependent transmission, harvesting can, in theory, cut transmission rate by lowering population density. Culling of red foxes (*Vulpes vulpes*) to lower transmission by reducing population density was used to limit rabies before vaccination was common [5]. Lowering the disease prevalence can also be achieved for diseases with close to frequency-dependent transmission, but is far more challenging. Harvesting shortens the duration of the infectious period, and can target spatial clusters [6], or high-risk demographic groups [7]. The hunting and culling of hosts remains a strategy of active mitigation, including the culling of badgers (*Meles meles*) to limit bovine tuberculosis [8,9], culling of hare (*Lepus timidus*) to limit louping ill virus [10] and culling of wild boar (*Sus scrofa*) to limit both tuberculosis [11] and African swine fever [12].

The efficacy of culling in controlling wildlife diseases is determined by complex ecological and epidemiological interactions [3]. The probability that an introduced pathogen persists in a population beyond the early phase of an outbreak is influenced by two stochastic processes occurring at the individual level: (i) the actual transmission of infection between an infected and susceptible individual, and (ii) an infected individual surviving natural and harvest mortality [13]. Intensive harvesting in the early stage of an outbreak will decrease the second probability, and, therefore, the likelihood of removal of all infected individuals increases. However, a quantitative evaluation of the probability of eliminating all infected individuals by selective harvesting strategies upon detection of disease at a very early epidemic stage is lacking. The relationship between harvest intensity and the removal of infected individuals is not necessarily linear, warranting a model-based approach. We hence modelled this simple principle of disease elimination to the case of chronic wasting disease (CWD) [6,14], a fatal prion disease of cervids [15,16]. CWD has adverse long-term impacts on deer populations [17,18], and continues to spread into new populations in the USA, Canada and, more recently, Norway. Harvesting is the main disease mitigation tool for CWD [14,19], but is challenging since most studies suggest a near-frequency-dependent or only weakly density-dependent transmission rate. The use of harvesting to limit CWD in the USA and Canada has typically dealt with the later endemic stage, and the focus has been on limiting transmission rates [7,20], using spatially targeted culling [6,21], or removing high-risk groups [22]. CWD is notoriously difficult to eliminate by harvest once established, due to environmental contamination of prions [19]. CWD prevalence in a population can remain low for several years before it becomes endemic [23]. In the early stage of a CWD outbreak, the main route of transmission is by direct animal-to-animal contact [24], and the removal of all infected hosts is likely to end a developing epidemic. The hunting of cervids is typically selective [25]. Adult males are more likely to be CWD infected than adult females in deer [26,27], and the selective harvesting of adult males would likely increase the probability of success in removing most of the infected individuals earlier compared to that of random harvesting.

Here, we used an agent-based model that incorporated individual-level heterogeneity to evaluate the efficacy of intensive harvest strategies in removing infected hosts from a population, especially intended for the early outbreak stage (low disease prevalence). We used CWD in reindeer (*Rangifer tarandus*) as our case study. In Norway, the first detection of CWD in 2016 elicited a full eradication of the Nordfjella population of over 2000 reindeer [28,29]. We obtained detailed data on all CWD cases and abundances for the entire Nordfjella population that was eradicated [30], allowing us to evaluate how different levels of intensive harvesting implemented in a realistic reindeer population can affect the removal of CWD-positive individuals without eradicating the population. Full host eradication would be more effective but is not always feasible due to other stakeholder concerns. Further, an adult male harvested on 3 September 2020 from the adjacent Hardangervidda reindeer population tested positive for CWD after more than 3500 negative tests (since 2016). Preliminary estimations of

CWD prevalence in the Hardangervidda population suggest a range of one to eleven infected adult males (H.V., unpublished results, 2020). Here, we calculated the probability of removing all CWD-infected individuals by culling massively at this early epidemic stage among reindeer in Hardangervidda, Norway. By how much can high harvesting rates increase the probability of removing all infected individuals depending on (i) true CWD prevalence and (ii) selective harvesting strategy? We calculate probabilities of removing all CWD-positive individuals depending on harvest strategy. Due to estimation uncertainties at low prevalence, we recommend using scenarios of CWD prevalence covering the 95% credible range of estimates rather than detailed disease dynamics models. The approach can also be used in an iterative, adaptive management fashion by estimating prevalence after each harvest season with updated data, and forecasting probabilities for full CWD removal as a basis for further harvesting. The approaches suggested here are also relevant for areas of North America with recent detections of CWD in deer populations, and for other diseases detected at early epidemic stages, particularly those with slow epidemic growth and a long incubation period. The approach is only useful at early disease stages, before environmental contamination is likely to affect the disease dynamics notably or cause re-emergence. The approach can even be used pre-emptively, before the disease is detected in a population.

# 2. Material and methods

## 2.1. Study areas

Our study areas consisted of the Hardangervidda and Nordfjella reindeer management areas (figure 1). Hardangervidda is a mountain plateau of around 8000 km$^2$ with a population consisting of over 9000 reindeer. Nordfjella consists of two management zones due to the intersection of a road, limiting connectivity. Nordfjella Zone 1 is approximately 2000 km$^2$, and CWD was first detected in the area in 2016. The population consisted of about 2000 reindeer before population eradication was completed by 1 May 2018. Nordfjella Zone 2 is situated between Hardangervidda and Nordfjella Zone 1, but currently with no detected cases of CWD and with some 500–600 wild alpine reindeer. Northeast of Nordfjella is the semi-domestic reindeer range of Filefjell. Connectivity to Filefjell is limited by a road, and after discovery of CWD, is also limited by a fence along the road [31].

## 2.2. Reindeer population data and CWD surveillance

In Norway, the population abundance of reindeer is estimated from a monitoring system coordinated by the Norwegian Institute for Nature Research. An established population estimation model integrates four annual censuses [32]. We used previously published data on the estimated mean population sizes of Hardangervidda reindeer for 2019 [33], which are available on GitHub [34]. The population in Nordfjella Zone 1 counted 2024 individuals at the time of eradication [28]. Due to some missing data on age and sex, we included 1982 out of the total population of 2024 in the simulations (table 1).

CWD surveillance is operated by the Norwegian Veterinary Institute and the Norwegian Institute for Nature Research [35]. During the ordinary hunting period in the fall of 2016 in Nordfjella, three reindeer infected with CWD were shot. In the eradication process, a total of 14 CWD-infected reindeer were found: one yearling male, nine adult males and four adult females [30]. We estimated the remaining number of CWD-infected individuals at the onset of eradication, 10 August 2017, and every month of the marksmen culling (1 Nov, 1 Dec, 1 Jan 2018, 1 Feb) and before the final stage of cull (15 Feb), using an established estimation model [30,36].

## 2.3. Management system using selective hunting quotas

We used harvesting strategies that reflected the quota system for reindeer in Norway [37]. These quotas are usually given as 'free licences', 'adult females, including yearlings of both sexes' and 'calves' (i.e. young of the year). Adult females, including yearlings of both sexes, are used because it is difficult to distinguish adult females from yearlings in the field, due to their similar body sizes and appearances. Recreational hunting in Norway is mainly for meat. Adult males have much larger body sizes than adult females, and many hunters also value the trophy of large males. 'Free licences' are usually expensive and are perceived by hunters as an adult male licence, so we used 'adult male' licences in the simulations. For Hardangervidda, there was an unusually large proportion of males as part of the

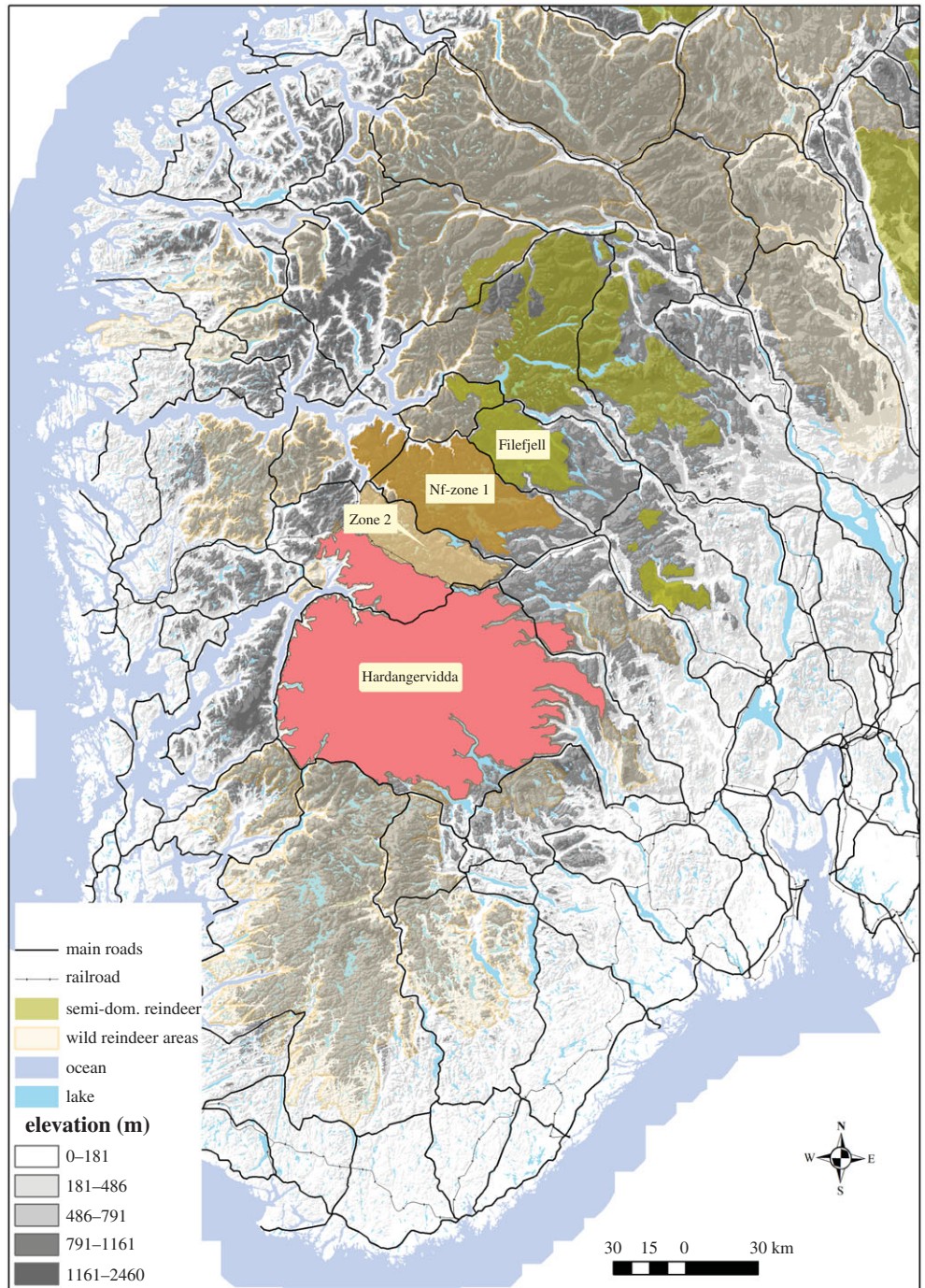

**Figure 1.** An overview of the reindeer populations with ongoing CWD management in southern Norway. Nf-zone 1, Nordfjella Zone 1. Zone 2, Nordfjella Zone 2.

CWD pre-emptive management in 2019 [33]. We, therefore, ran simulations for the standing population before the harvest in 2019 (with a more even sex ratio).

## 2.4. Model used for simulations of harvesting strategies

We used the established agent-based model MO*Ov*POP*surveillance* that was initially developed for evaluating CWD surveillance in a white-tailed deer (*Odocoileus virginianus*) population in Missouri, USA [38]. Since we used a simplified version of an earlier published model [38], we only highlight the

**Table 1.** An overview of the pre- and post-harvest population sizes of reindeer in the Nordfjella (Zone 1) population, Norway, given different harvesting strategies. Harvest rates of yearlings (both sexes) follow adult females (Hf). Hc, Harvest rates of calves. Adult sex ratio, adult m : f.

| parameter | pre-harvest | baseline | strategy 1 | strategy 2 | strategy 3 | strategy 4 | strategy 5 | strategy 6 | strategy 7 |
|---|---|---|---|---|---|---|---|---|---|
| **harvest rates (%)** | | | | | | | | | |
| Hm | | 20 | 70 | 90 | 100 | 100 | 100 | 100 | 100 |
| Hf | | 20 | 50 | 60 | 70 | 80 | 90 | 95 | 98 |
| Hc | | 13 | 13 | 24 | 40 | 60 | 80 | 95 | 95 |
| **population size (*n*)** | pre-harvest | post-harvest | | | | | | | |
| adult males | 540 | 432 | 163 | 53 | 0 | 0 | 0 | 0 | 0 |
| adult females | 753 | 602 | 381 | 302 | 227 | 151 | 74 | 38 | 15 |
| yearling males | 144 | 116 | 72 | 57 | 44 | 30 | 15 | 8 | 3 |
| yearling females | 141 | 114 | 70 | 55 | 42 | 28 | 14 | 7 | 3 |
| calves | 404 | 352 | 350 | 310 | 243 | 162 | 81 | 20 | 20 |
| population size | 1982 | 1616 | 1036 | 777 | 556 | 371 | 184 | 73 | 41 |
| adult sex ratio | 1 : 1.4 | 1 : 1.4 | 1 : 2.3 | 1 : 5.7 | NA | NA | NA | NA | NA |
| prob. all CWD + removed | 0 | 0 | 0 | 0.01 | 0.01 | 0.15 | 0.52 | 0.81 | 0.88 |

main points here. We give a more complete overview in electronic supplementary material, with the standard Overview, Design concepts, and Details (ODD) protocol for agent-based models [39,40].

The reindeer in this study are nomadic with no clear home range behaviour, and there was no spatial clustering of the CWD cases within the Nordfjella range [41]. Hence, spatial heterogeneity was not included for reindeer, which differs from the situation of CWD in the population of white-tailed deer. The model is coded in the open-source Java-based modelling environment NetLogo [42] and is freely available in the digital repository CoMSES Net Computational Model Library [43]. An important feature of this model is the ability to simulate age–sex-specific harvest scenarios in a realistic host population with relevant host characteristics incorporated in the model programmes. We parametrized the model using demographic and harvest data for the reindeer populations and evaluated alternate harvest strategies by performing virtual experiments, that is, simulations. The harvest in each age–sex class was simulated as a random process. For each harvest scenario, we undertook 100 iterations of the model to assess the probability of eliminating CWD-positive individuals (mean number of CWD-positive individuals removed from the population, and the proportion of iterations where 50, 75 and 100% of CWD-positive individuals were removed from the population). We also documented the post-harvest population composition for each harvest scenario.

## 2.5. Simulations of hunting strategies given CWD prevalence (scenarios)

We ran simulations of different harvesting strategies for the Nordfjella population with known exact CWD prevalence and abundances, and for the Hardangervidda population with estimated population sizes and CWD prevalence as scenarios.

### 2.5.1. Hunting strategies

We used alternative quotas: 'adult males' (greater than or equal to 2.5 years old), 'adult females (including yearlings)' and 'calves'. CWD prevalence in calves was extremely low, but we harvested a sufficient number of calves to limit extensive orphaning. We fixed a similar harvest rate of adult females and yearlings to reflect the quota system. We ran a series of hunting strategies varying the adult male and female harvest rates. The baseline consisted of ordinary harvest rates and the composition within the range of the observed empirical harvest rates from the affected areas. We then ran strategies with increasingly heavy harvesting rates, up to the full eradication of all adult males (table 1).

### 2.5.2. CWD prevalence

For Nordfjella, we had information on the exact population composition (table 1) and the apparent CWD prevalence since this population was fully eradicated [30]. We made two observations of how many CWD-infected individuals were removed by harvesting:

— Nordfjella ordinary hunting in 2016: two adult males, one adult female.
— Nordfjella eradication in 2017/2018: one yearling male, nine adult males, four adult females.

For Hardangervidda, the CWD prevalence was uncertain, partly due to only a few samples of female reindeer. It may have been as low as only one or two adult males with a 95% upper limit of 11 adult males (H.V., unpublished results, 2020). However, there were more adult females in the population (table 2), so it cannot be ruled out that there could be up to 20 infected reindeer. We, therefore, considered the following scenarios for the number of CWD-infected deer based partly on the range of the initial estimation of CWD prevalence and possible CWD growth if action was delayed:

— Scenario 1: two adult males,
— Scenario 2: three adult males, one adult female,
— Scenario 3: nine adult males, three adult females, one yearling,
— Scenario 4: 18 adult males, six adult females, one yearling.

**Table 2.** An overview of the pre- and post-harvest population sizes of reindeer in the Hardangervidda population, Norway, given different harvesting strategies. Harvest rates of yearlings (both sexes) follow adult females (Hf). Hc, harvest rates of calves. Adult sex ratio, adult m : f.

| parameter | baseline | | strategy 1 | strategy 2 | strategy 3 | strategy 4 | strategy 5 | strategy 6 | strategy 7 |
|---|---|---|---|---|---|---|---|---|---|
| **harvest rates (%)** | | | | | | | | | |
| Hm | 20 | | 70 | 90 | 100 | 100 | 100 | 100 | 100 |
| Hf | 20 | | 50 | 60 | 70 | 80 | 90 | 95 | 98 |
| Hc | 13 | | 13 | 24 | 40 | 60 | 80 | 95 | 95 |
| **population size (n)** | pre-harvest | post-harvest | | | | | | | |
| adult males | 2356 | 1883 | 706 | 237 | 0 | 0 | 0 | 0 | 0 |
| adult females | 3376 | 2700 | 1694 | 1348 | 1005 | 674 | 339 | 169 | 67 |
| yearling males | 665 | 532 | 331 | 266 | 202 | 133 | 67 | 33 | 13 |
| yearling females | 723 | 578 | 358 | 289 | 218 | 144 | 72 | 38 | 15 |
| calves | 1286 | 1121 | 1120 | 978 | 770 | 516 | 259 | 64 | 64 |
| population size | 8406 | 6814 | 4209 | 3118 | 2195 | 1467 | 737 | 304 | 159 |
| adult sex ratio | 1 : 1.43 | 1 : 1.43 | 1 : 2.40 | 1 : 5.7 | NA | NA | NA | NA | NA |
| **probability all CWD + removed** | | | | | | | | | |
| Scenario 1 | 0.03 | | 0.53 | 0.8 | 1 | 1 | 1 | 1 | 1 |
| Scenario 2 | 0.01 | | 0.18 | 0.45 | 0.76 | 0.75 | 0.88 | 0.91 | 0.98 |
| Scenario 3 | 0 | | 0.01 | 0.06 | 0.17 | 0.34 | 0.65 | 0.83 | 0.94 |
| Scenario 4 | 0 | | 0 | 0.01 | 0.1 | 0.28 | 0.46 | 0.72 | 0.86 |

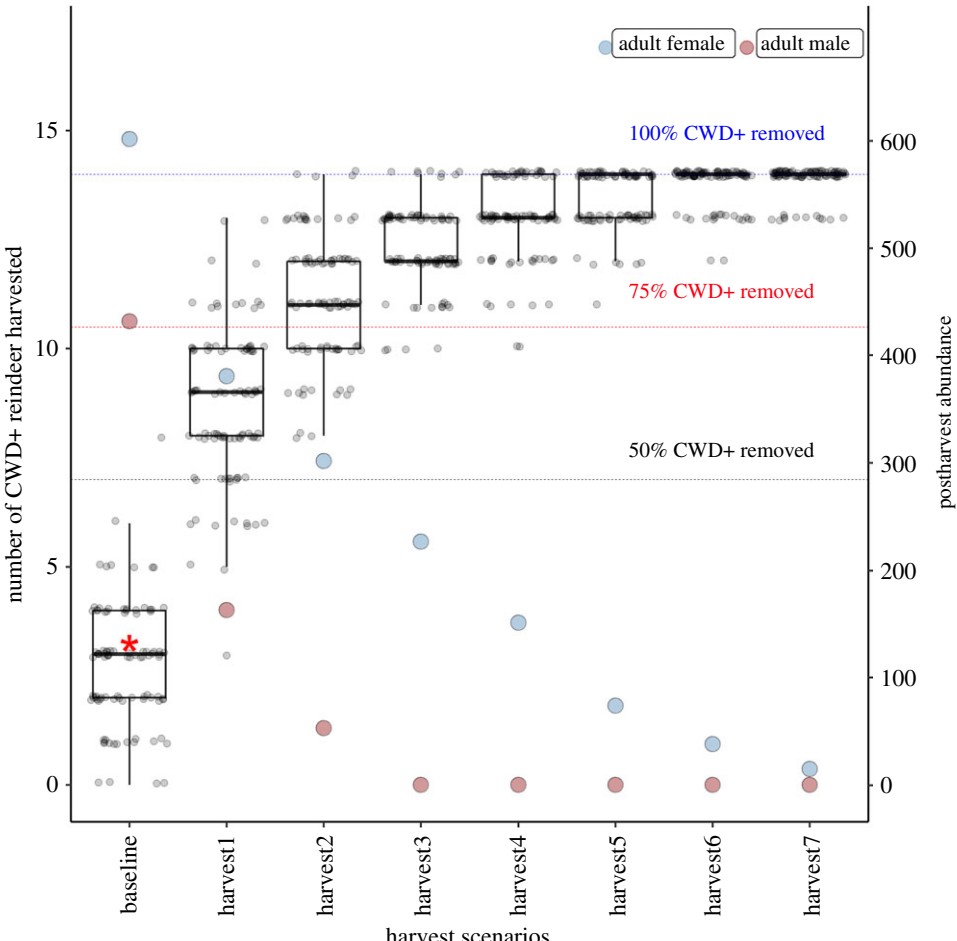

**Figure 2.** The likelihood of removal of CWD infected individuals from the Nordfjella population with 14 known CWD cases given different harvesting strategies. The baseline was within the range of ordinary harvesting rates used in these populations, which in 2016 removed three CWD-positive reindeer (red asterisk). Each point represents one of 100 simulations. The boxplots show the distribution in the boxes as 25%, 50% and 75% quartiles, with the whiskers at 5% and 95% quartiles. The right axis gives post-harvest population sizes for adult males (red) and females (blue).

## 3. Results

### 3.1. Nordfjella with known CWD prevalence

For the Nordfjella population of 1982 individuals with 14 observed CWD cases, ordinary harvest rates and composition removed less than 50% of the infected deer in most simulations (figure 2). With the ordinary hunting practices of 2016, only three CWD-positive reindeer were removed, which was very close to the theoretical expectation. Using harvesting rates in the range of 50–90% for both adult males and females frequently removed 50–75% of the CWD-infected individuals, but rarely ever 100%. Only extreme harvesting rates of 100% adult males and 95–98% adult females had some chance to remove all CWD-infected reindeer but reduced the population size to below 100 individuals (table 1). In a full eradication using a non-selective harvesting strategy, the last CWD-infected individual was removed when the remaining population counted 183 individuals (figure 3).

### 3.2. Hardangervidda—scenarios for CWD prevalence

The probability of CWD elimination from Hardangervidda depended on both the harvesting regime, in terms of composition and rates, and heavily on the initial CWD prevalence (table 2 and figure 4). For Scenario 1 with only two adult males infected, only one out of 100 runs with ordinary harvesting pressure succeeded in removing both males, while harvesting 90% was usually successful in removing both males without detriment to the female population. In Scenario 2, both of the two adult males

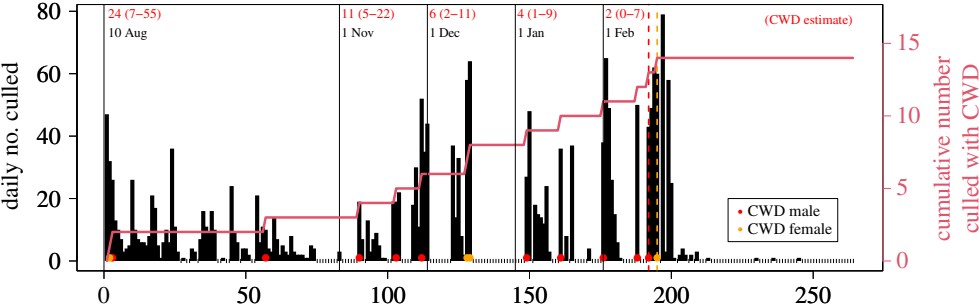

**Figure 3.** The timeline from the onset of eradication and the actual dates when the 10 males (one yearling, nine adults) and four females (all adults) that were CWD positive were removed from the Nordfjella reindeer population, Norway. There were 183 remaining healthy individuals when the last CWD-infected animal was culled. The cumulative number of removed CWD-infected individuals is in red, and the estimated number of remaining infected individuals is presented as numbers at the onset of each month with 95% credible interval in brackets.

**Figure 4.** The likelihood of removal of CWD infected individuals from the Hardangervidda population given different harvesting strategies and CWD prevalence scenarios. (*a*) Scenario 1 with two CWD-infected adult males. (*b*) Scenario 2 with three adult males and one adult female that were CWD infected. (*c*) Scenario 3 with nine adult males, three adult females and one yearling male that were CWD infected. (*d*) Scenario 4 with 18 adult males, six adult females and one yearling male that were CWD infected.

and the single infected female were also removed with heavy male harvesting and moderate reduction of the female population size. Scenarios 3 (13 infected individuals) and 4 (25 infected individuals) included a markedly higher CWD prevalence, and the success of removing all infected animals was low unless harvesting rates were extreme. Only the full removal of adult males and greater than 80% removal of the adult female population had a fair probability of removing all infected individuals. This would lead to post-harvest population sizes well below 1000 individuals (table 2).

# 4. Discussion

We have simulated how ordinary harvesting regimes and more extreme selective harvesting affects the probability of eliminating all infected individuals in populations with a recently detected disease. Our approach provides one avenue to eliminate wildlife diseases if detected sufficiently early without the eradication of the entire host population. This is the CWD-management situation for reindeer in Norway, and also for many areas of the USA and Canada, as CWD continues to expand its distribution range. We highlighted the benefits of using selective harvesting in these efforts, but the elimination method has a number of potential pitfalls. Unfortunately, the time window for success was incredibly short, and massively invasive harvesting was required already with only 15–25 infected individuals. There was no clear guarantee of success even when combating CWD at the early epidemic stages. Further, we highlighted how large variations in harvesting rates, and hence sample sizes, is a challenge for surveillance programmes and may lead to an 'estimation crisis'. Hence, the method has clear limitations and has potentially adverse population impacts.

## 4.1. Iterative adaptive management facing uncertain prevalence

At early epidemic disease stages, the growth of the disease can be stochastic and impossible to estimate with precision [13]. In the specific case of CWD, stochasticity may also cause deviations from the assumed demographic pattern of prevalence (prevalence ratios: 1/2 in yearlings = 1 in adult females = 3 in adult males) [30]. In addition, empirical data on CWD growth in populations of reindeer are lacking. Therefore, we deliberately did not include disease dynamics in the model, and instead varied the disease prevalence in the scenarios. The major limitation of the elimination method is uncertainty regarding whether or not the last infected deer has been removed for a given harvest. Nevertheless, as culling and testing proceeds, management can update models to at least come closer to a realistic target for harvest (figure 3). Before the onset of host eradication in Nordfjella, the estimated number of CWD-infected reindeer was 24 with wide 95% credible intervals (7–55), while the actual detected number was 14 infected individuals [30]. In February 2018, our model predicted two (95% credible interval: 0–7) remaining CWD-infected individuals, while the true value was four. Very close to the end of culling, on 15 February 2018, the model predicted two (95% credible interval: 0–5) remaining infected individuals, and the observed value was also two. However, this could reflect a theoretical optimum, as the estimation model was built on this dataset. In a real situation, the population size is not known precisely. By chance, 183 reindeer remained at the time when the last CWD-infected reindeer was removed (figure 3).

The first detection of CWD in an adult male in the Hardangervidda population was partly due to the extreme male-biased harvest implemented for early disease detection [33]. These heavy harvesting rates enable a precise estimation of CWD prevalence in adult males. However, the low harvest of females and yearlings requires future harvests that include a considerable number of females and yearlings to improve CWD prevalence estimations upon the detection of cases, or requires the obtainment of increased evidence for the absence of CWD in this population segment. Hence, an iterative, adaptive management schedule is recommended [44]. Adaptive management of CWD using harvesting is not a new idea [14], and it appears a natural choice of strategy. Delaying action too much would lower the probability of success due to the slow, but anticipated, growth of CWD. By contrast, adaptive management can provide updated prevalence data and it may be easier to implement politically and logistically. CWD has a slow initial growth; thus, in a real-world situation, it would be easier to implement a continued, heavy harvesting over time, to increase the probability of a disease die-out, rather than one extreme harvest, as simulated here. Even if culling does not remove all individuals, it may nevertheless increase the likelihood of a disease die-out due to stochastic natural mortality and transmission rates.

## 4.2. 'Estimation capital', 'estimation crisis' and potential for 'super-spreading'

Harvesting aids in combating disease, but also in the estimation of disease in the case of CWD and other diseases often relying on post-mortem diagnostics [36]. Large natural fluctuations in host populations of fecund, short-lived species have been shown to undermine the surveillance of wildlife diseases [45]. We warn that hugely variable harvest rates of long-lived species will yield similar problems in surveillance. When initiating culling, one aims to harvest more than the recruitment rate to the population. This heavy harvesting yields considerable sample sizes and hence enables accurate estimation of disease prevalence. We refer to this as the 'estimation capital' at the onset of culling. During culling, the removal of infected individuals will cut off transmission, lowering the growth of disease. However, this situation of heavy harvesting is unsustainable in the long term. After a period of unsustainable harvesting to combat disease, harvesting intensity must be relieved for the population to recover. This will yield a situation later with a limited harvest and hence low sample sizes, which we term an 'estimation crisis', as one will no longer be able to measure disease prevalence or reliably establish the likelihood of the absence of disease. If infected individuals remain, they will now live longer due to lower harvest rates and infect more individuals compared to that of a more constant harvest pressure, raising the potential for the 'super-spreading' of disease. At low population sizes, only harvesting adult males would provide at least half of a solution to this problem in the case of CWD, as they have a higher likelihood of infection and harvesting of adult males of polygynous species usually do not affect the population growth rate [33], as yearling males would likely impregnate females. Clearly, non-invasive methods, for example, disease detection in faeces [46], would be of enormous benefit to determine whether all CWD-infected individuals have been removed. A current limitation of these sensitive methods is instances of false positives, but even if this caused extra removal of some herds, it may pay off in the end. If the population becomes very small, one may also consider using invasive strategies like 'test and cull' to overcome the uncertainty about disease status [47].

## 4.3. Limitations, adverse effects and selective harvesting regime

The tactics of full host eradication as completed in the Nordfjella Zone 1 reindeer population has the benefits of knowing that all infected individuals were eradicated with certainty, but then restocking the area becomes a problematic process [48]. A benefit of the proposed elimination method is that it does not require the difficult and expensive process of restocking a given area; however, there are several other limitations with this method. We assumed that there was no environmental contamination that could cause reinfection, which is a crucial point for success. The epidemiological role of environmental transmission is regarded as low in early epidemic stages [24], but even a single CWD carcass or infected faecal matter can initiate a new epidemic [49]. Removal of offal and other leftovers from any CWD-infected deer is crucial. While full host eradication is extremely invasive, the potential for adverse impacts with the 'knock out' method may also be substantial. A large proportion of the female population, and all adult males, must be removed to have a chance of being successful in CWD Scenarios 3–4 (figure 4c,d). Fewer adult males in the population may delay calving [50], culling can stress the animals, reducing body conditions and there is a risk of extensive culling leading to further geographical spread of disease [51]. All of these limitations should be considered in the local context. In the case of reindeer, the loss of genetic variation also becomes an issue when harvesting a population to a low level [52]. The general recommendation is to retain an effective population size (Ne) of 500 or 1000 individuals to retain evolutionary potential [53]. The reindeer populations of Nordfjella and Hardangervidda have already lost a considerable part of the genetic heritage due to gene flow from semi-domestic reindeer [54]. Other parts of the southern metapopulation of reindeer in Norway may be required for genetic rescue if further genetic losses occur due to genetic drift if a population is kept at low numbers for long periods. In the case of white-tailed deer, spatially targeted harvests on CWD clusters by marksmen are used to limit disease prevalence [6,21]. Similarly, heavy culling of white-tailed deer was used in Minnesota to avoid the establishment of a wildlife reservoir for bovine tuberculosis [55]. Other disease systems, such as an emerging fungal pathogen in amphibians, also involve host culling [56,57]. Using our approach in a more confined area would have a less detrimental impact than removing the whole population. A limitation for developing less adverse hunting strategies to combat CWD is the lack of a good understanding of the transmission dynamics of CWD under different field conditions and contexts [19].

# 5. Conclusion

Unplanned 'crisis management' of wildlife diseases may do more harm than good [58]. We have previously advocated the use of simulations to guide wildlife disease management, particularly in early epidemic stages, due to the difficulty of detecting infected animals when the prevalence is very low [13]. We regard it beyond the scope of our article to consider the logistical difficulties in achieving such harvesting, both politically and practically [4]. The Norwegian authorities will decide on which tactics are feasible for the Hardangervidda population, counting 8000–10 000 reindeer, given the various socioeconomic and political (public resistance) and conservation biological (loss of genetic diversity) concerns. Here, we provided a basis for an alternative response to full host eradication in the case of the early detection of serious wildlife diseases. We highlight that this is solely from a probability-of-disease-elimination perspective, not including the substantial adverse impacts.

Ethics. No animals were harvested as part of this study.

Data accessibility. Data and relevant code for this research work are stored in GitHub: https://github.com/anyadoc/ABMDataNorwayReindeerCWD and have been archived within the Zenodo repository: https://doi.org/10.5281/zenodo.4501249 [59].

Authors' contributions. A.M. conceived the main initial idea and drafted the paper. The idea and draft were further developed together with A.B. and H.V. A.B. conducted the simulations. H.V. estimated CWD prevalence in Nordfjella. C.M.R. provided CWD and reindeer data and knowledge. All authors edited the subsequent drafts and approved the final version.

Competing interests. The authors declare no competing interests.

Funding. Own funding.

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
