## [Peer Review File · Royal Society Open Science]

Review History

Decision letter (RSOS-210124.R0)

Dear Dr Mysterud:

It is a pleasure to accept your manuscript entitled "Harvest strategies for the elimination of low prevalence wildlife diseases" in its current form for publication in Royal Society Open Science.

At this stage, we ask that you please archive your GitHub code within the Zenodo repository: <https://guides.github.com/activities/citable-code/>. By doing this, a formal, citable DOI will be associated with your data record, and an open license (CC-BY preferred) can be applied to your data. We would then ask that you please update your data availability statement to read as:

Reports © 2021 The Reviewers; Decision Letters © 2021 The Reviewers and Editors;
Responses © 2021 The Reviewers, Editors and Authors. Published by the Royal Society under the terms of the Creative Commons Attribution License <http://creativecommons.org/licenses/by/4.0/>, which permits unrestricted use, provided the original author and source are credited

"Data and relevant code for this research work are stored in GitHub: [GitHub URL here] and have been archived within the Zenodo repository: <https://doi.org/zenodo.....> [ref number].

You can expect to receive a proof of your article in the near future. Please contact the editorial office (openscience@royalsociety.org) and the production office (openscience_proofs@royalsociety.org) to let us know if you are likely to be away from e-mail contact – if you are going to be away, please nominate a co-author (if available) to manage the proofing process, and ensure they are copied into your email to the journal.

on behalf of Dr Krijn Paaijmans (Associate Editor) and Professor Pete Smith (Subject Editor).

Associate Editor Dr Krijn Paaijmans Comments to Author:

Associate Editor
Comments to the Author:
The authors adequately addressed the main concerns from two previous reviewers. The manuscript is well-written, and the conclusions sound.
